

**Modeling and observing the Lake Albano dynamics**
Authors:
**Anita Grezio[1], Damiano Delrosso[1], Marco Anzidei[2], Marco Bianucci[3], Giovanni**
**Chiodini[1], Antonio Costa[1], Antonio Guarnieri[1], Marina Locritani[4], Silvia Merlino[3],**
**Filippo Muccini[4], Marco Paterni[3], Dmitri Rouwet[1], Giancarlo Tamburello[1], Georg**
**Umgiesser[5,6]**
[1] Istituto Nazionale di Geofisica e Vulcanologia, Bologna, Italy
[2] Istituto Nazionale di Geofisica e Vulcanologia, Roma, Italy
[3] CNR, Lerici, Italy
[4] Istituto Nazionale di Geofisica e Vulcanologia, Roma2, Italy
[5] ISMAR-CNR, Venezia, Italy
[6] Klaipeda University, Klaipeda, Lithuania
Corresponding author: anita.grezio@ingv.it, https://orcid.org/0000-0001-6848-7589
**Abstract**
Lake Albano is a monomictic volcanic crater lake in Central Italy with $CO_2$-rich waters
presenting $CO_2$ concentration varying over time. Depending on the period of the year, the
lake is characterized by strong stratification or rather overturning events. In the warm season,
the heating of the surface water results in a highly stratified vertical density profile, while in
the cold season, the surface water cooling leads to a potential vertical instability of the water
column. In this case, a partial/deep overturning of the lake water column may occur with the
degassing in the atmosphere of the $CO_2$ which was accumulated as dissolved species in the
deep water layers following seismically induced gas recharge, months to years before. Such a
process has been periodically observed in Lake Albano in the past and could pose a potential
hazard to the surrounding environment and population. A 3D numerical model is
implemented to investigate the lake dynamics and the occurrence of overturning events. The
model is validated and calibrated using both historical observations and measurements
acquired during this study. These include temperature and salinity profiles from the deepest
central portion of the lake, surface water temperature time series recorded by sensors installed
on the lake shores, mounted on remotely operated vehicles, and on low-cost, innovative, self-
powered drifting buoys. The latter have also been used to assess the modeled surface
circulation of the lake.



## 1. Introduction

After the limnic eruption of Lake Nyos in 1986 (Kanari 1989; Kling et al. 1989), volcanic lakes have been recognized as a rare but devastating source of disasters (Kusakabe 2017, Kling et al. 2015). Limnic eruptions are caused by the accumulation of magmatic $CO_2$ in non-acidic crater lakes, reaching supersaturation, or triggered by an external factor (e.g. earthquake, landslide, strong winds). The gas recharge might occur for two reasons: a) a sudden injection through the lake bottom of a relevant quantity of $CO_2$, or b) a high $CO_2$ concentration built up within the lake for a long time (Rouwet et al. 2019).

Italy hosts twelve volcanic lakes (Albano, Nemi, Averno, Lucrino, the two Monticchio lakes, Bolsena, Bracciano, Vico, Mezzano, Martignano, Monterosi, Specchio di Venere-Pantelleria, Telese) with different physical and chemical characteristics (Cioni et al. 2003; Chiodini et al. 2000; Chiodini et al. 2004; Stoch et al. 2007). The Albano maar is the deepest among the volcanic crater lakes of Italy, being about 167 m deep (in 2007, see Anzidei et al., 2007). It is the youngest of the monogenetic and polygenetic phreatomagmatic craters located along the northern and the western slopes of the Colli Albani volcanic complex (De Rita et al.1987; De Rita et al.1988; Trigila 1995; Villa et al.1999; Funiciello et al. 2003; Marra et al. 2003; Freda et al. 2006; Giordano et al. 2010). The lake has a long history of level changes and catastrophic events, which started with the formation of the Albano crater ~70 ka B.P., and continued during pre-historical times. Geological and historical evidence suggests that a large overflow of the lake occurred in 396 B.C.E. due to a rapid increase of the water level. The event contributed to fill the valleys on the north flank of the Albano maar crater forming the Tavolato di Ciampino, an area characterized by a flat topography which is presently the site of the international airport (Funiciello et al. 2003). In Lake Albano, a water overturning or a mixing of deep and shallow waters could bring $CO_2$ from the bottom of the lake to the surface with a potentially hazardous release of $CO_2$ (Funiciello et al. 2003; Carapezza et al. 2008; Chiodini et al. 2012). Such overturns may occur when the equilibrium of the water-column stratification is modified by water density variations. The potential risk of Lake Albano (20 km southeast of the centre of Rome) is due to exposed elements (people presence, economic and touristic activities). As such, to estimate the potential gas hazard of Lake Albano, numerical modeling of the lake water dynamics is crucial for understanding its current and future behavior and stability.

In 1989, Lake Albano was affected by a large $CO_2$ input pulse during a seismic swarm below Colli Albani volcano. On the basis of historical literature, at least two similar anomalous degassing events took place between 1829 and 1927, when five seismic crises occurred (Rouwet et al. 2019). A recent (August 2020), short seismic swarm resulted in a minor $CO_2$ recharge in deep water layers (Rouwet et al. in prep.). Apart from those significant episodes, a moderate degassing of likely magmatic origin is present at irregular intervals. Lake Albano is considered a monomictic lake and almost every winter, during the water overturning (**Figure 1**) the lake commonly releases $CO_2$ (Chiodini et al. 2012) in non-hazardous amounts, thereby preventing long-term $CO_2$ accumulation in the bottom waters. This is in contrast to the dynamics of tropical stratified lakes, such as Lake Nyos (Rouwet et al. 2021), where lake overturn does not occur and gas build-up can thus occur to eventually reach CO2



supersaturation conditions, followed by a sudden gas burst.
In this study, we investigate the characteristics of lake stratification and overturning events at
Lake Albano through the results of 3D numerical model simulations, supported by
instrumental data collected from temperature sensors attached to drifters and buoys, used to
calibrate and validate the model.

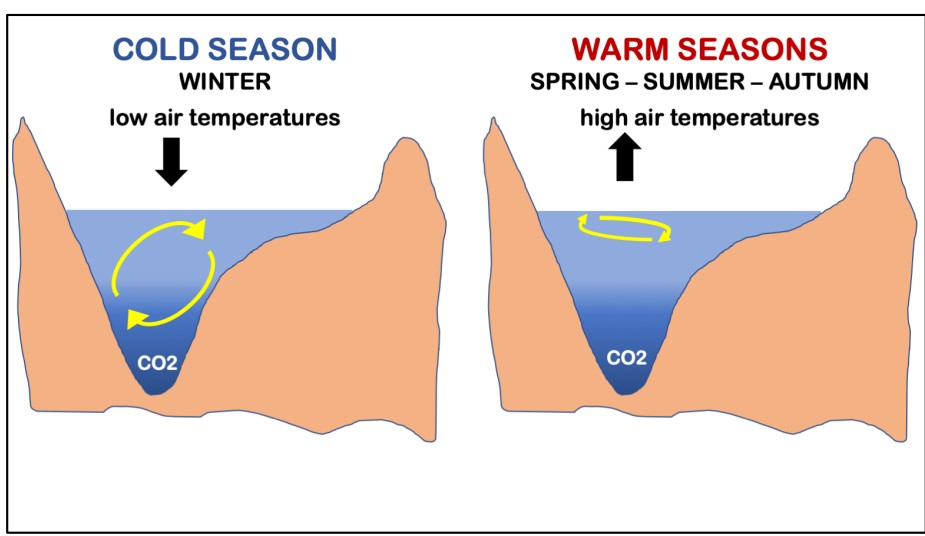


**Figure 1**: *Schematic illustration of the lake seasonal conditions with a typical winter*
*overturning and summer stratification (depth and horizontal extension of the lake are not in*
*scale).*

**2. Modeling the volcanic Lake Albano**
In this study, the physics of the lake dynamics is investigated for the first time for Lake
Albano by a numerical model. It was not explored by previous works (Carapezza et al. 2008;
Chiodini et al. 2012; Rouwet et al. 2021) because the usual description of the lake behavior
was provided by analyzing vertical profiles of the physical-chemical properties along the
vertical profile at maximum lake depth. This 2D description may be adequate for a general
characterization of the lake conditions, but a 3D representation is essential for fully resolving
the lake dynamics and variations of physical-chemical characteristics in time and space.
The general ocean circulation numerical model SHYFEM (System of HydrodYnamic Finite
Element Modules) (Umgiesser et al. 2004) is implemented for Lake Albano in order to (1)
reproduce the lake dynamics throughout the year, and (2) given its importance in hazard
evaluation, represent the volcanic lake system during the winter overturning.
The volcanic lake system is subjected to recurrent phases of recharging and emission of $CO_2$.
The yearly release of $CO_2$ depends on the vertical stratification, which is determined by the
lake surface-atmosphere heat fluxes. Thus, the implementation of a numerical model forced



by atmospheric reanalysis fields enables the representation of seasonal and interannual
cycles.

**2.1 System of HydrodYnamic Finite Element Modules (SHYFEM)**

A configuration of the SHYFEM model was implemented for Lake Albano (**Figure 2**).
SHYFEM is a framework of numerical models (SHYFEM, http://www.ismar.cnr.it/shyfem
and https://github.com/georgu/shyfemcm-ismar) to simulate the water movement and
temperature and salinity variables in an aquatic environment. It is a community model
developed by Italian and international institutes. SHYFEM consists of a finite element 3-D
hydrodynamic model, a transport and diffusion model and a radiation transfer model of heat at
the water surface. SHYFEM was previously successfully applied to many coastal environments
(Ferrarin and Umgiesser 2005; Ferrarin et al. 2010; Bellafiore et al. 2011; De Pascalis et al.
2011; Ferrarin et al. 2013, Umgiesser et al. 2014) and lakes (De Pascalis et al., 2009; Le Thi et
al. 2012). For more details of the model equations and their solution, please see Umgiesser et
al. (2004).

**2.2 Lake Albano model settings and parameterizations**

The lake bathymetry is derived from Anzidei et al. (2006; 2007; 2010), originally provided at
2 m of resolution. The model grid horizontal resolution ranges from 34 m to 66 m, due to the
varying size of the grid elements (**Figure 2**). The grid consists of 3016 nodes, 5831 elements,
and 89 vertical z levels, with thickness variable from 1 m (from the surface to the depth of 50
m) to 3 m (from 50 m to the maximum depth of 167 m). This horizontal and vertical
discretization appears adequate to satisfactorily resolve the major horizontal dynamical
structures and the vertical stratification of the lake.
No slip condition is set at the boundary/bottom ($u$ and $v$ velocity components equal 0). At the
boundary, water fluxes could be activated to account for sources such as groundwater
contributions.  However, despite qualitative indications on the locations of inflow/outflow
(Mazza et al. 2015), the absence of data throughout the year prevents the inclusion of
groundwater flux values in the model used in the present study.

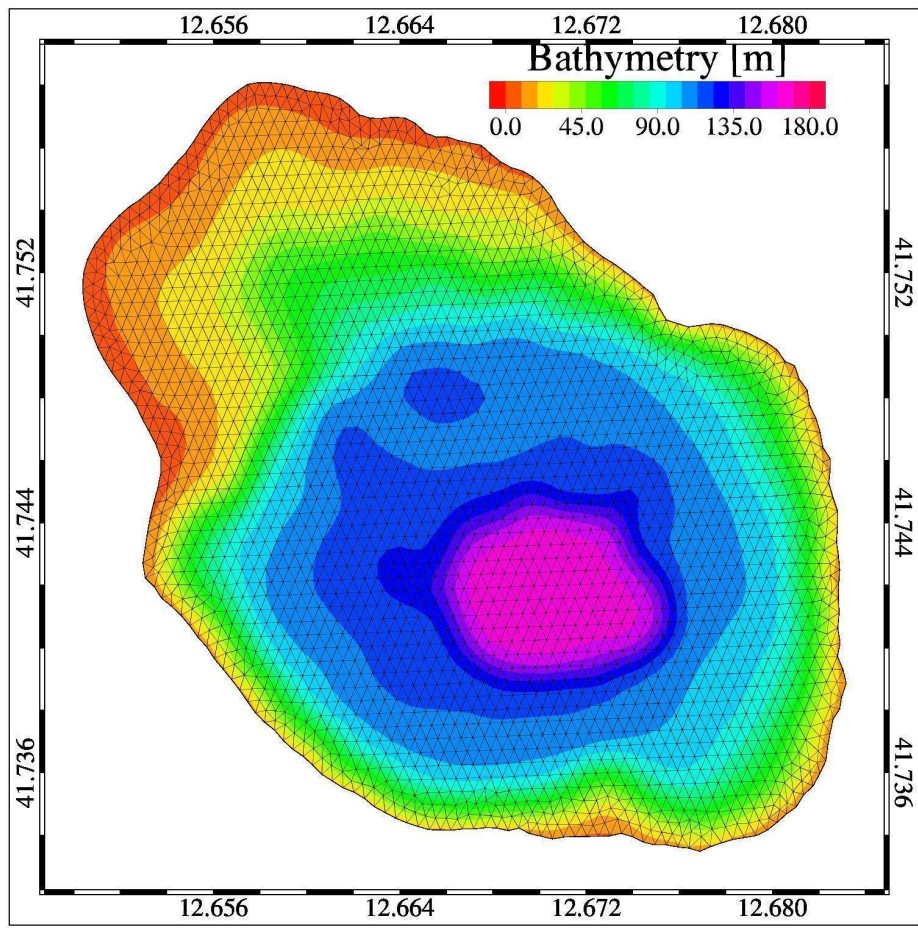


**Figure 2**: *Model bathymetry of Lake Albano with the model mesh superimposed.*

The model starts from rest. The initial conditions are based on homogeneous temperature (*T)*
and salinity (*S*) fields derived from averaging the vertical profiles of background observations
available from previous studies (Cioni et al. 2003; Carapezza et al. 2008; Ellwood et al.
2009). The temperature and salinity observed data were collected along the vertical profile of
the deepest part of Lake Albano. A set of monthly temperature measurements was available,
so the initial *T* field is based on the time-averaged observed values for the month used to start
the simulation, in this case January (**Figure 3a**). On the contrary, the paucity of salinity data
(main components are *Na, K, Mg, Ca, HCO$_3$, SO$_4$, Cl*), allowed only the setting of a mean
annual *S* field profile (**Figure 3b**). In order to construct the 3D initial conditions fields
required by the model, the temperature and salinity profiles are replicated across all the points
of the model grid.





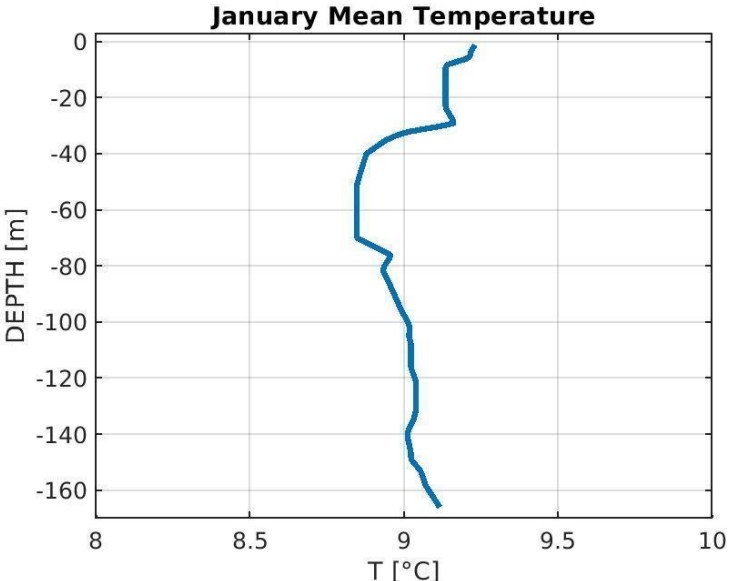

145                                                                                          *a)*

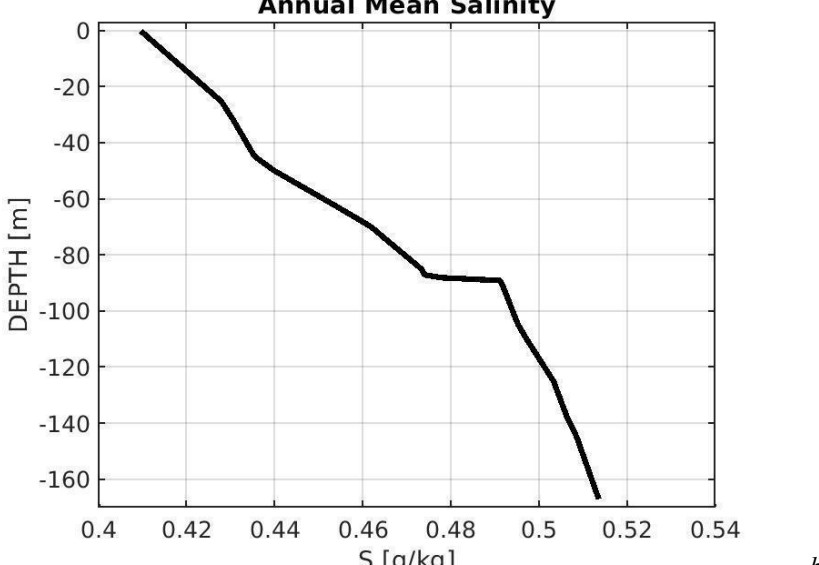

146                                                                                          *b)*

**Figure 3**: *a) Initial Conditions for temperature (T) and b) salinity (S) fields derived from observations available in the deepest part of Lake Albano.*


The atmospheric fields forcing used in the numerical model (10 m zonal and meridional wind
components, 2 m dew point temperature, mean sea level pressure, 2 m temperature, total
precipitation, total cloud cover, surface solar radiation downwards) are derived from the



ERA5 reanalyses product (Hersbach et al. 2020; Bell et al. 2021), which has a 1/4-degree
horizontal resolution and an hourly temporal resolution.
In order to exclude bias and trends in the numerical solution due to atmospheric forcing, a set
of perpetual year experiments is performed. The simulations are forced by ERA5 reanalysis
fields averaged over the period 1979 - 2019 to build a climatological forcing data set.
Two numerical experiments were performed with the aim of
1) reproducing the seasonal thermocline according to the available observations, and
2) avoiding trends and biases that could cause deviations from the observed vertical
temperature profiles after several years, especially in the deepest levels.
In the case of the numerical experiment 1 the tested model parameter was the vertical
diffusivity, while in the case of the numerical experiment 2 the tested model parameters were
related to the air-water fluxes bulk formulas and were tested in absence of precipitation input.
The results of the numerical experiments showed that the vertical diffusivity value which
optimizes the agreement between the modeled temperature profiles and the available
observations is $1.0\times10^{-5}$. This setting in the numerical experiment 1 maintains the thermocline
shallower than 40 m depth during the year (**Figure 4)**.
The heat flux formulation that minimizes the temperature trends over the course of the
simulation is the one proposed by Large and Pond (1981), used also in Princeton Ocean Model
(POM). This setting in the numerical experiment 2 avoids trends and keeps the temperature
variation below 0.05 °C (well below the observed monthly mean variability) along the vertical
profiles. The interannual differences of the temperature profiles in the deepest part of the lake
are calculated month by month and are shown in **Figure 5**.





**Figure 4**: *Numerical experiment 1 for different vertical diffusivity values used to maintain the seasonal thermocline below 40 m depth.*







**Figure 5**: *Numerical experiment 2 for air-water fluxes bulk formulas as in the Princeton Ocean Model used to reduce temperature deviation from observed data over time.*

## 3. Model validation and observations





After the testing phase shown in the previous section, which was devoted to identifying the
optimal model setup, a numerical simulation over the period 1st January 2020 - 31st December
2023, was performed. To monitor the thermodynamic characteristics of the lake and support
the validation activities of the implemented numerical model, in the framework of the
MACMAP project (*https://progetti.ingv.it/it/progetti-dipartimentali/ambiente/macmap*) data
loggers at fixed locations (since May 2022) and drifters deployed into the waters of the lake
(August 2022 campaign) were used to collect temperature data and information on surface
circulation, respectively.
**3.1 Eulerian data**
Two temperature loggers (Tinytag Aquatic 2, Gemini TG-4100, resolution ± 0.01 °C) were
installed in May 2022 on the southeastern and western shores of the lake to continuously
record the surface water temperature. Measurements were collected every 30 minutes, stored
and retrieved during field campaigns (**Figure 6**).

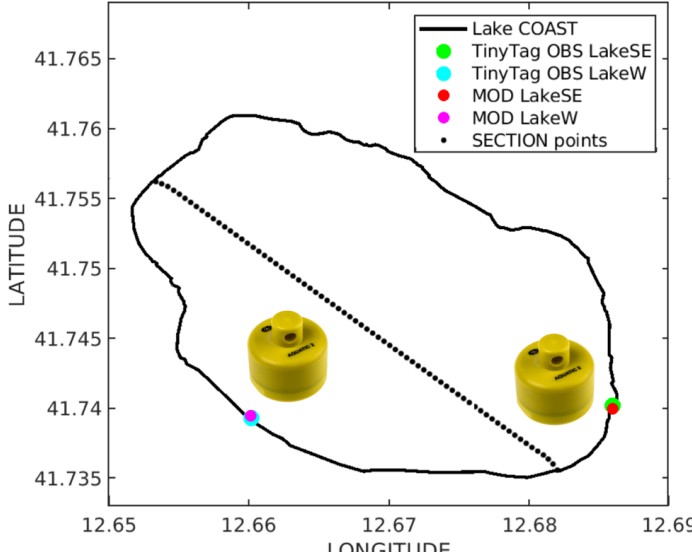


**Figure 6**: *Locations of the installed temperature loggers (Tinytag Aquatic 2, Gemini*
*TG-4100, resolution ± 0.01 °C, https://it.rs-*
*online.com/web/p/datalogger/0283915?cm_mmc=IT-PLA-DS3A-_-google-_-*
*CSS_IT_IT_PMAX_Catch+All_NEW-_--_-*
*283915&matchtype=&&gclsrc=aw.ds&gad_source=1&gbraid=0AAAAADkeWNNt2RVFl68*
*WQ1l1dWetSAEo9&gclid=CjwKCAiA5eC9BhAuEiwA3CKwQswZ0Q5C0R1ey9OHN6ur1PP*



*aN_9m92kM0DvESYGS9zfyw-5d6pNkXBoCf7QQAvD_BwE ) and the nearest model grid*
*points. In addition, the black dots indicate the section shown in Figures 10-13.*

The recorded time series were used to assess the model's capability in reproducing the lake
temperature and its variability throughout the year. The average root mean square error
(RMSE) between modeled and observed temperature is slightly larger than 3 °C. However, in
the winter time the RMSE is reduced. In fact, in the temporal interval 01-Jan-2023 00:30:00-
31-Jan-2023 00:30:00 the RMSE is 2.12. The temperature seasonal variability is correctly
reproduced by the model, as confirmed by **Figure 7,** although an evident bias is present with
respect to the data recorded by the data logger installed on the southeastern shore of the lake
(blue curve in **Figure 7**). It is worth to note that the southeastern Tinytag data logger
probably moved and emerged from the lake waters in 2022, but it was not possible to
reconstruct the exact time window of this unplanned displacement.

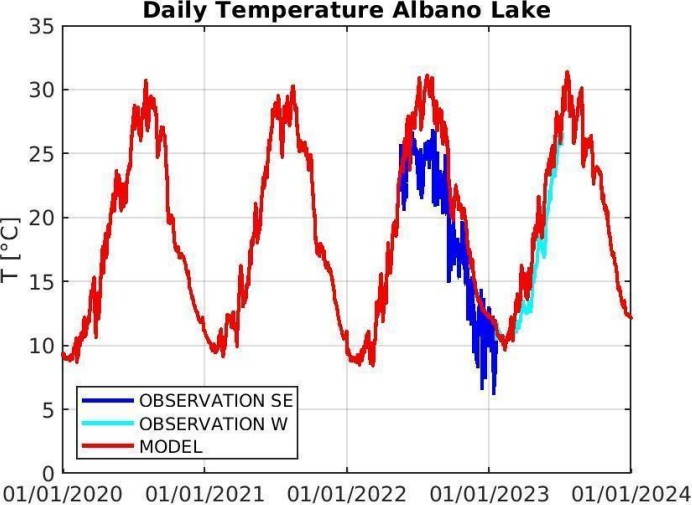


**Figure 7**: *Temperature recorded by the data loggers (blue and cyan curves indicate the SE*
*and W observations, respectively) and modeled temperature at the nearest locations where*
*the data loggers (Tinytags) were installed (red curve).*

**3.2 Lagrangian data**
The lake's surface circulation was investigated using in-house assembled, low-cost drifters
originally employed to study the dynamics of litter of riverine origin (as described in Merlino
et al. 2023). The initial drifters (http://carthe.org, Novelli et al. 2017) were entirely
transformed for the purpose of this project. The innovative, reliable, robust, self-powered and
low cost "Marine Litter Trackers" (MLT) were developed in the framework of the ML-DAR



project (A multidisciplinary method to study the Marine Litter Dispersion from the Arno
River mouth: a study case to support citizen science, funded by INGV) and finally they were
re-adapted to be utilized as surface current trackers for this study. Two different kinds of
support were used: wooden tablets and floating supports. Some types of drifters were
equipped with cloth drogues that allow them to better follow the surface current and be less
affected by wind at the lake surface. Moreover, temperature sensors were installed on the
devices, in order to acquire surface water temperature data, in addition to tracking surface
currents.

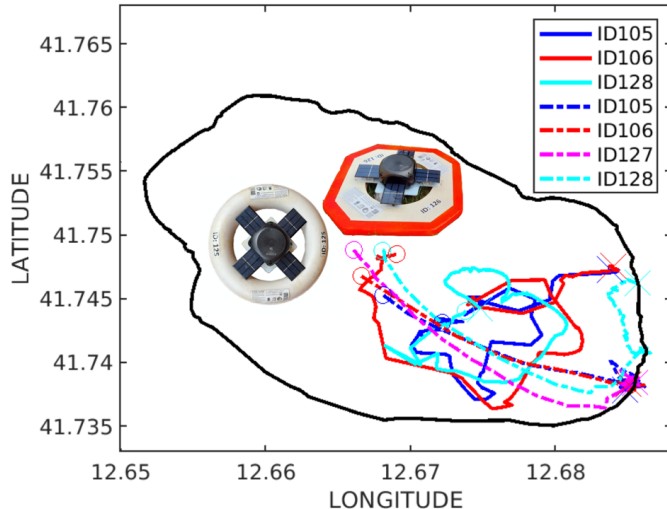


**Figure 8**: *Images of the types of drifter deployed in Lake Albano and trajectories acquired by*
*the instruments (identified by the ID numbers) during the second survey (13-15 May 2023).*
*Solid lines indicate the trajectories of the drifters deployed on day 13th, dashed lines are the*
*trajectories of the same drifters deployed on day 14th. Picture of the drifters used, including*
*the modified CARTHEs, from which the battery packs and satellite antenna were removed,*
*replaced by the small central box with the consumer electronics of our mini drifters (SD card,*
*batteries, GPS antenna, GSM antenna), from the 4 photovoltaic cells in a halo, and to which*
*a temperature sensor was added (Merlino et al. 2023, DATA SET-repository: https://data-*
*nautilos-h2020.eu/erddap/tabledap/mini_drifter.html).*

The drifters were deployed in Lake Albano during two different surveys: a first one carried
out on 30th August and 1st September 2022 (mainly devoted to testing the drifters in a lake





environment) and a second one from 13th to 15th May 2023 (the acquired trajectories are
shown in **Figure 8**). A comparison was then performed between the trajectories recorded by
the drifters and the modeled surface currents for the three days of the second survey. A first
numerical experiment, forced by ERA5 atmospheric reanalyses, did not yield satisfactory
results, with drifter trajectories following a cyclonic pattern and modeled surface currents an
anticyclonic one in the same time frames (not shown). The reason for this discrepancy could
be due to the coarse, for our study area, spatial resolution of the ERA5 dataset (0.25 degree),
which probably limits the possibility to correctly reproduce the local wind regime. For this
reason, another numerical experiment was performed, retrieving the wind components from
an anemometer station located a few kilometers north-east of the lake
(*https://www.wunderground.com/dashboard/pws/IMARIN65/graph/2023-05-15/2023-05-*
*15/daily*). In this case a better agreement between the modeled surface currents and the drifter
trajectories is observed (**Figure 9**). Some discrepancies may be due to the course model
resolution which is not able to catch small features that drifters may be capable to follow.

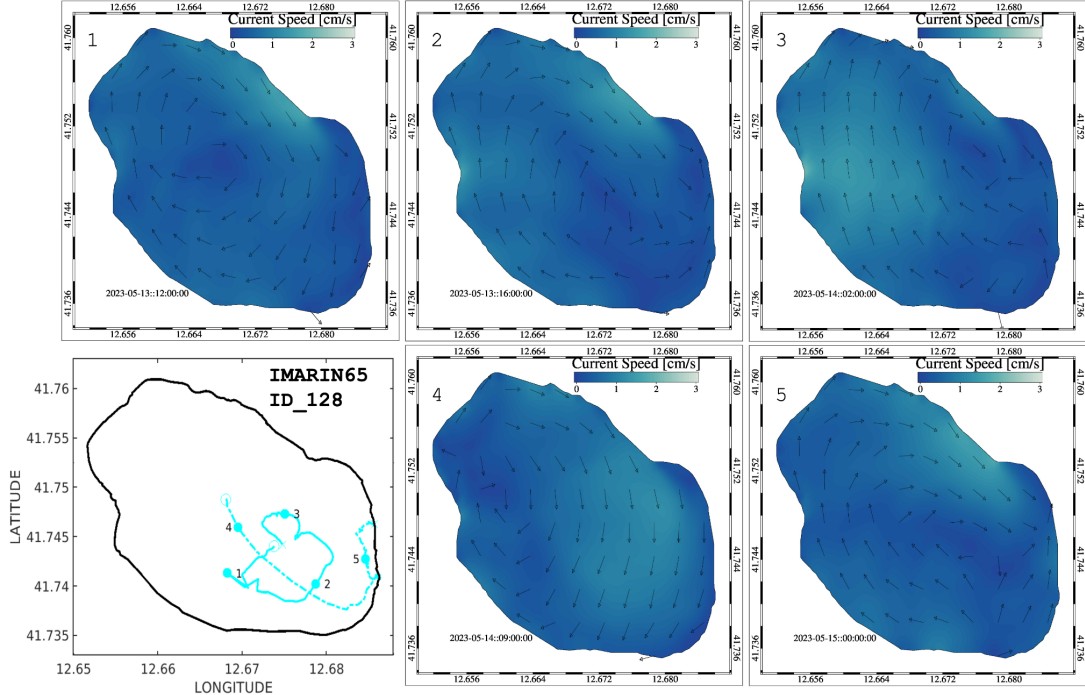



**Figure 9**: *Modeled surface currents on 13-15 May 2023 using the wind velocity field from the*
*anemometer IMARIN65 (model snapshots shown in the 1-5 panels) and trajectory of the*
*drifter ID_128 (first lower left panel).*

**3.3 Winter Overturning**



The model simulation covers a four-year period (01/01/2020 - 31/12/2023), with analyses
conducted on both a seasonal and interannual basis. **Figures 10-13** show temperature
longitudinal section for the winter period (January, February, and March, although some deep
vertical water temperature variations occur also in April). From the beginning of spring
(April) to the beginning of autumn (October), a remarkable vertical stratification is present
(**Figure 14** and **Supplementary Materials**), while from November a vertical mixing that
deepens progressively is observable from the temperature profiles.
Spatial variations in water density are basically driven by temperature, since salinity does not
show significant temporal and spatial variability (see **Figure 3**). The winter overturning is
strongly correlated with vertical temperature variations. Among the four years of simulation,
two years (2020 and 2022) exhibit winter overturning, which begins in the second decade of
January. Colder, and hence denser, water formation occurs in the northern shallow part of the
lake. Then, due to the lake's bathymetric configuration, denser waters move towards the deep
central part of the lake. Later, in February, cold water formation also occurs at the center of
Lake Albano (**Supplementary Materials**).
On some days in February and March it is possible to notice cold water patches moving from
the deeper layers towards the surface (**Supplementary Materials**).
In some cases, intermediate layers (around 75-80 m depth, see video in Supplementary
material), exhibit colder temperatures compared to the layers directly above and below, until
the bottom, after the winter overturning. In these cases, vertical stability is still maintained,
because it is the total density that determines the stratification: despite being slightly warmer,
the deepest water layers are more saline, resulting in a higher total density at depth.



**Figure 10**: *Modeled temperature T [°C] longitudinal sections in winter 2020.*




**Figure 11**: *Modeled temperature T [°C] longitudinal sections in winter 2021.*







**Figure 12**: *Modeled temperature T [°C] longitudinal sections in winter 2022.*




**Figure 13**: *Modeled temperature longitudinal T [°C] sections in winter 2023.*





304
 **Figure 14**: *Monthly averaged modeled and observed temperature profiles for the years 2020-*
*2023 (no observations available for October).*

## 3.4 Overturning Index and Stability Index

In order to quantify the potential occurrence of overturning events, we computed the
overturning depth $D_{overt}$



$$D_{overt} = \sum_{i=L_s}^{L_b} d_{level}(i) \cdot I_x(i)$$

where $L_s = 1\,m$ and $L_b = 167\,m$ are the surface and the bottom levels, $d_{level}$ is the depth of
each single level in the model, and $I_x$ is the overturning index defined as following:
$$I_x(i) = 0, \quad if \quad T_{level}(i) \geq T_{mean}$$
$$I_x(i) = 1, \quad if \quad T_{level}(i) < T_{mean}$$
with the mean temperature $T_{mean} = mean\,(T_{layerII}, T_{L_b})$. The $T_{layerII}$ and $T_{L_b}$ are,
respectively, the water temperature at the top of *layer II* (defined in Chiodini et al. 2012)
which is located below 95 m, and the temperature of the deepest layers.
In other words, the index $I_x$ compares the surface temperature with the temperature at
maximum lake depth, so when surface temperatures are lower than deep temperatures,
overturning conditions may be present. **Figure 15** shows the Howmoller diagram of the
temperature profiles at the deepest point of the lake for the entire simulation period. As such,
the levels involved in the overturning mechanism can extend below 100 m, reaching depths
of up to 120 m, hence deeper than previously thought (Chiodini et al. 2012).

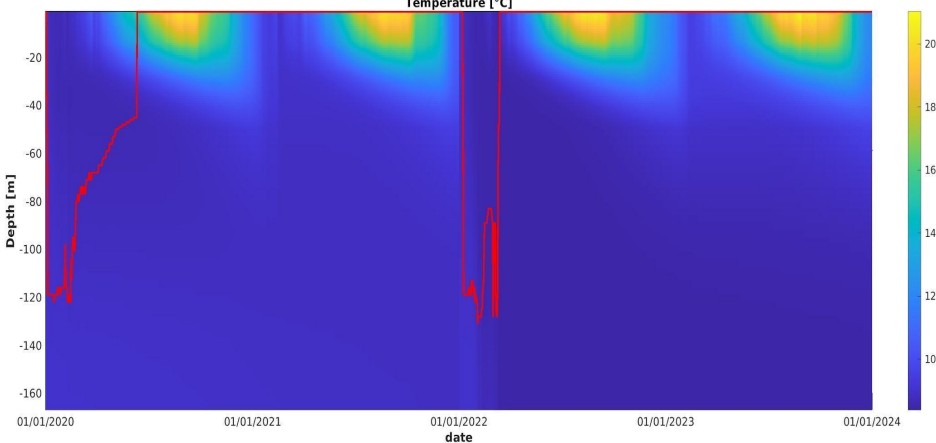


**Figure 15**: *Howmoller diagram of the temperature profiles at the deepest point of the lake*
*and $I_x$ index values (red line).*

By comparing temperature values at different lake depths, the results show that temperatures
at 1 m (T1) are lower than those at depths below 95 m for 35 days in the 2020 winter and 29
days in the 2022 winter at the point of maximum lake depth. This time window indicates
*when* and *for how long* the surface water is colder than the deeper layers, making it able to
sink and trigger lake overturning. In 2021 and 2023, by contrast, surface temperatures (T1)
never fall below the temperature at 40 m.



For completeness, the Schmidt stability index (Schmidt, 1928) was computed to characterize
the stability of a stratified water column. It is a measure of the energy required to completely
mix a stratified lake with an arbitrary vertical density distribution, taking into account the
volume of the lake basin (Kirillin and Shatwell, 2016). It represents the work required for its
mechanical mixing without heat exchange with the environment per unit area (Smirnov et al.,
2024). The Schmidt stability index $St$ is expressed by
$$St = g\, A_0 \int_0^{Hmax} (z - z_v)\, \rho_z\, A_z\, dz$$

where $g$ is acceleration of gravity, $\rho$ is water density at depth $z$, $A_0$ is lake surface area, $A_z$
is lake area under the isobath $z$, $Hmax$ is maximal lake depth, $z$ is the depth to the center of
lake volume, calculated as: $z = \frac{1}{V} \int_0^{Hmax} (z - z_v)\, A_z\, dz$, $V$ being the lake volume. The
adequate assessment of Schmidt stability requires eliminating variations in temperature
profiles, so model daily averaged temperature values are used in the analysis. The computed
St values (shown in **Figure 16**) fall within a range consistent with recent studies on the
physical limnology of Italian lakes (Ambrosetti et al. 2002) and confirm that the energy
required to vertically mix the water column is lower for 2020 and 2022.

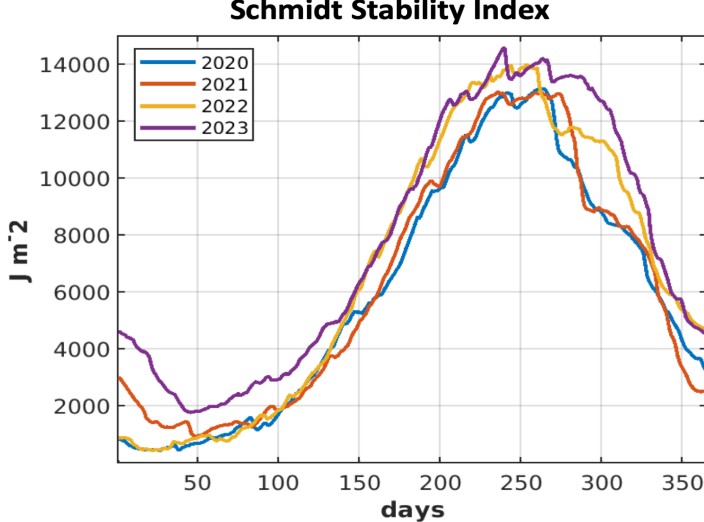


**Figure 16**: *Schmidt stability index computed for the years 2020-2023.*

**4. Discussion and Conclusions**
The 3D SHYFEM model was implemented to simulate the physical dynamics of the volcanic



Lake Albano. A numerical experiment was performed from 01/01/2020 to 31/12/2023 for
investigating seasonal and interannual lake variability.
Major findings of this study are:
- the winter overturning mechanism is driven by atmospheric forcing and does not occur
every year, but only when the surface thermal conditions are suitable. Atmospheric
forcing causes the cooling of the lake surface and triggers the overturning, typically in
January. Over the four years simulation, winter overturning occurred twice (in 2020
and 2022), starting after mid-January and lasting for a few weeks, before the spring-
summer stratification resumes until the following winter. In both 2021 and 2023, the
stratification starts very early and surface cooling results insufficient for the lake to
overturn.
- cold water formation begins in the shallow northern part of the lake and spreads
southeastward toward the deeper lake levels. When surface cooling is intense in the
middle of the winter, the cold water sinks down to the lake bottom at 167 m;
- the winter overturning in 2020 and 2022 involves the rise of water layers from depths
greater than 95 m, with cold water patches moving towards the surface. Given that (i)
$CO_2$ is more soluble in cold water, (ii) deep water layers are enriched in $CO_2$, and (iii)
no bubble degassing is observed during winter overturn, this upward movement of cold
$CO_2$-enriched deep water may be the physical mechanism behind $CO_2$ release during
winter overturning, in agreement with previous degassing models and trends
(Carapezza et al. 2008; Chiodini et al. 2012; Rouwet et al. 2021);
- after the winter overturning, surface temperatures follow the seasonal variations of the
atmospheric temperature. Strong thermal stratification of the lake persists above the
thermocline during the warmest seasons. Below 40-50 m, the temperature usually
remains between 8.0-9.5 °C throughout the year, as confirmed by the available
observations.
Although the spatial resolution of ERA5 atmospheric fields may be considered too coarse for
the lake area, the air-lake heat fluxes in this region are considered suitable for investigating the
winter overturning mechanism. In fact, the modeled surface temperature follows the observed
seasonal temperature variations. To better reproduce surface circulation patterns, local wind
forcing would be required, in order to introduce finer-scale variability.
Another limitation is the lack of groundwater data. Even if the SHYFEM model can simulate
water fluxes at the lake boundaries, the lack of this data prevents the inclusion of interactions
between groundwater and lake water, as well as processes related to further temperature
variations. Including such processes would require a multi-scale approach that combines
multiple measurement methods, considerable constraints and uncertainties, and the estimation
of the fluxes between groundwater and lake water at different spatial and temporal scales.
Increasing surface water temperatures due to global warming could enhance vertical water
stratification and potentially inhibit or reduce the frequency of the overturning process. As a



consequence, in the case of less frequent winter overturning, $CO_2$ accumulated over the years
(following seismically induced recharge, Chiodini et al. 2012), may increase limnic gas hazard
for Lake Albano when overturn does occur. In the extreme case where Lake Albano ceases to
overturn due to ongoing atmospheric warming, $CO_2$ could remain stored in the deep lake layers
until supersaturation conditions will be reached, potentially triggering a "Nyos-type" gas burst.
So, the numerical model implemented in this study, tracking atmospheric and surface water
temperatures at Lake Albano, could provide a basic framework for monitoring future degassing
dynamics. Moreover, simulations of Lake Albano could reproduce the past dynamical
evolution of the lake and may also reconstruct its behaviour in relation to both the seismic
swarms in the Colli Albano area in 1987-1990 and the decline of the lake water surface levels
due to anthropic activities over the last decades, which is expected to continue in the coming
years. Concerning the fluctuation of the lake level, a continuous lowering of its level started in
1970 (Anzidei et al. 2010) with a continuous acceleration during the following decades (Capelli
et al. 2000; Capelli and Mazza 2005; Riguzzi et al. 2008; Mazza and Capelli 2010). The lake
level remained about stable between 1940 and 1960. Then, a lake level fall occurred at a mean
rate of 8.8 cm/yr in 1960-2005 and then at 20 cm/yr in 1990-1997 (Anzidei and Esposito 2010).
The cause of the lake level fall has been largely attributed to the excessive ground water
withdrawal (Capelli and Mazza 2005) or in connection with shallow seismicity (Bianchi et al.
2008; Chiarabba et al. 2010) and ground uplift (Amato and Chiarabba 1995; Riguzzi et al.
2009; Anzidei et al. 2010). The hypocenters of the latest earthquakes occurred in 1987-1990 in
this area were aligned along a NW-SE striking structure across the Lake Albano and the other
craters of the Colli Albano volcano (Amato et al. 1994, Bianchi et al. 2008). It is worth noting,
that during this seismic period, a significant lake level drop occurred in 1990. This phenomenon
has been addressed to an increased permeability of the lake basin in response of endogenous
processes.
Finally, despite the lake level changes, the model implemented could be applied to other
volcanic lakes to investigate their dynamics and associated physical and chemical processes.

**Author Contribution**
AG conceived the study, implemented the model, computed the simulations and wrote the
draft manuscript; DD prepared the forcing and supported the model implementation for
parallel computing; GU supported the model implementation; MA provided the Albano
bathymetry; DR installed the tinytags and prepared the Eulerian data; MB, SM, MP, ML and
FM deployed and prepared the drifters providing the lagrangian data; GT prepared the wind
velocity field from the anemometer. All authors worked collaboratively to interpret the data
and finalized the writing of the manuscript for publication.

**Competing Interests**
The authors declare that they have no conflict of interest.



**Acknowledgments**
We thank Claudia Fratianni for supporting the procedure to include the "Supplementary
Materials" in the ERDDAP system for the easier access to scientific data.
This study has been developed in the framework of the MACMAP project funded by Istituto
Nazionale di Geofisica e Vulcanologia (Environment Department). Bathymetric Surveys
were performed in 2007 under the Colli Albani Project V-3 funded by the Italian
Dipartimento della Protezione Civile.

**Supplementary Materials**
Additional materials are movies of the simulation of the January, February, March and April
months for the 2020-2023 years and they can be downloaded at the following link
http://oceano.bo.ingv.it/erddap/files/albano_lake_model_video/ .

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
