# Peer review of "Modeling and observing the Lake Albano dynamics"

_EGUsphere, 2025_

## Author Comment (AC1)

Dear Editor,

Thanks for your letter and your editorial support. Also, we wish to thank the two anonymous referees whose comments allowed us to make changes and checks for improving the manuscript. In the following paragraphs we write our answers to the *referees' comments* point by point.

Best regards.

Anita Grezio and co-authors.

**Reviewer RC1**
*In their manuscript, the authors investigate the overturn and*
*hence the possible sudden release of dissolved CO2 to the atmosphere. They*
*present measured data and data from a 3D numerical model;*
*The investigation covers 4 years. Those differ in the depth of the deep mixing.*
*The authors present a big package of work. Doing both, field*
*measurements and running a numerical model requires a lot of*
*effort. The general understanding of processes is reported correctly.*

We thank the reviewer for recognizing our efforts in the implementation of the model and in gathering instrumental data in order to describe the Lake Albano processes reported in this study.

*The tools that are chosen are sort of suited for the purpose, though*
*not necessarily the ideal approach. The biggest shortcoming lies*
*in the fact that the goal of the investigation is not fully supported*
*by the results and I would even - looking at the results - interpret*
*things differently. (There is no good display of the mixing depth;*
*Howmoller is questionable; Schmidt number not suited and inconsistent*
*_ see below)*

We would greatly appreciate the opportunity to read the reviewer's interpretation of our results and to take their comments into careful consideration, in order to improve the manuscript. To address the reviewer's comments, we investigated alternative methods for analysing the results of our numerical experiments. In fact, firstly we moved the Figure 8 and we added a new Figure 9 in section 3.1 showing the monthly mean salinity profiles; and secondly we recalculated the mixing depth (see the paragraph 3.3 in the new version of the paper) and evaluated the density vertical structure.
In Fig. 8, the observed temperature profiles (black curves) are shown. We separate

the observed data in "OBS " retrieved from the literature referring to years different from those simulated and in "data1" acquired during the project for the period 2020-2023. Similarly, in Fig. 9 we provide observational and modelled data for salinity. The model results are represented by monthly means rather than time-specific profiles and we believe that the main features are reasonably well captured. We believe that the model is able to reasonably reproduce the vertical structure of the water column, because the temperature and salinity errors are within the seasonal variability.

*There are shortcomings in the Figures:*
*Firstly, Figures (4), (5), 6, 7, 10, 11, 12, 13, 14, (15)*
*contain text, axes values, legends that are not readable (those in*
*brackets "hardly readable"): If this text contains information for the reader, larger*
*fonts are needed. If there is no information, remove the text.*

The fonts in figures 4-5 and the other figures are changed to be more readable, following the reviewer's recommendations. For the other figures we think the font size is adequate.

*Secondly, the captions are not acceptable. Figures must be complete in themselves: looking at the graphics and reading the caption should suffice to understand what is displayed.*

The captions are changed, according to the reviewer's suggestions.

*Thirdly, the figures 10, 11, 12, 13: I understand there is lots of*
*work going into producing such displays: Hence I can relate to the fact*
*that the authors want to show their efforts: I agree, one or two of those panels make sense, but in general there is no longitudinal information: showing profiles would make more sense, and would indicate temperature gradients better than the colour coding.*

In our opinion, section figures provide a large amount of information which is different from those provided by the profiles. Profiles are effective in showing monthly/seasonal changes at representative points of the lake, and they are shown in Figures 4 and 5 for the model implementation, while Figure 8 and 9 assess the agreement between model and observations for Temperature and Salinity profiles. The section plots, which encompasses the entire lake from the shallow parts to the opposite site, make clear where and when the deep water formation occurs and how it spreads during winter. We show only a figure (Figures 13/14) as an example of the overturning process in the section. Moreover, considering the bathymetry, the

deepest part of the lake can be approximated as a paraboloid, implying that the information is likely symmetric along the longitudinal axis.

To provide additional insight into temperature gradients, as the referee required, we show the figure below which presents temperature profiles for selected days in January and February of the year 2020, confirming the evaluations on the sections.

[Figure]

*Fourth, Figure 16: something is wrong! each year should start with a value similar to the end of the previous year: the end of 2021 does not ft with the beginning of 2022.*

We checked the code and provided a new figure (values were not read properly in the code).

*The equations (no numbers) on page 20 are not understandable and probably formally not correct.*
*The major conclusion of the manuscript claims that deep circulation happened in 2020 and 2022, but not in 2021 nor in 2023. While I can relate to the*

*observation of variability, looking at the*
*displays in Fig 10, I would say the lake has not mixed deeper than 95 m*
*in 2020. In 2022 (Figure 12) I can see that an overturn may have happened.*
*I agree: no indication of this in 2021 nor 2023.*

In the equations on page 20 in the previous version of the paper, there were inaccuracies in the description of a variable that likely affected the comprehensibility of the equation: we defined $D_{overt}$ as the sum of the depth of each layer, whereas it is the thickness of each layer. The word "depth" is thus not appropriate, so the word "depth" must be changed in "thickness". Also, we added the equation numbers in the revised version of the paper.
However, we decided to change the approach and use another standard formula for the estimation of the mixed layer depth in section 3.3 in the new version of the paper.

*The Schmidt number is not really suited for an indication of mixing depth.*

We agree that the Schmidt Stability Index does not provide a precise measurement of mixing depth, but it offers valuable information about the lake's stratification by indicating whether the water column is more stably stratified or more susceptible to mixing. This index is a relevant indicator of stratification dynamics, as used in several studies and can be useful for future comparison between present time and future scenarios.

*The Howmoller approach delivers other results than the optical*
*impression of the displays. Do the authors state clearly, whether they*
*think Howmoller is a useful approach?*

The Hovmöller diagram is a standard tool used in climatological and meteorological studies to highlight patterns and structures. We believe that it is an effective way to visualize the spatial and temporal evolution of a variable, in this case the lake's vertical thermal or density structure, in the new version of the paper. We agree that in some cases a vertical profile may allow for a clearer observation, as we showed in figures 8 and 9.

*Usually mixing depths can be detected (1) in measurements: by oxygen*
*profiles, better even CO2 profiles as the release of this gas was the major intention*
*of this study (2) in numerical simulations: mixing depth*
*can be detected by inserting particles in the deep water.*

Regarding point (1): a concomitant paper on the CO2 observations and degassing events (Rouwet et al. 2025, personal communication) within the same project

indicates those events occurring in 2021, 2022 and 2023. For this reason we also investigated the salinity profiles and the density fields (Fig. 9 and Fig. 16). Regarding point (2): the suggested modelling requires additional efforts and is not implemented yet in the Lake Albano model, but we will take it into consideration for future developments. However, to provide more precise insights into the mixing depth, we computed the mixing depth using three different methods: the temperature threshold method (e.g. Wilhelm and Adrian, 2007), the density threshold method (e.g. Winder et al., 2009) and the maximum temperature gradient (e.g. Wilhelm and Adrian, 2007). Several different threshold values have been used in various studies, as described in Gray et al. (2020), for the first two mentioned methods. For the temperature threshold method we tested the corresponding threshold values indicated in Table 1 of Gray et al. (2020), while for the density threshold method (where the density has been computed following Chen and Millero, 1986) we tested the threshold values indicated in Figure 2 (M1 method) of Wilson et al. (2020). Finally, the mixed layer depth was estimated by density gradient from the surface, following Wilson et al. (2020) and the recomputed mixed layer depth was added in Fig. 16.

By the comparison our study with the $CO_2$ study from Rouwet et al, 2025 (in preparation) it became evident that: in 2020 the overturning was deep (as seen by temperature analysis showed in the previous version of the paper) but not so deep to be sufficient to produce a degassing event; in winter 2021, 2022 and 2023 the degassing events occurred. So, by considering the temperature analysis we missed information on other deep overturning events. Instead, by considering both the $CO_2$ observations (in Rouwet et al, 2025 in preparation) and the re-computed mixed layer depth using the density threshold method (Wilson et al., 2020) we finally found 3 deep overturning events concomitant with the degassing events in 2021, 2022 and 2023 (in agreement with the $CO_2$ observations and the model mixed layer depth).

Additional references (included in the paper):
- Wilhelm, S., & Adrian, R. (2007). Impact of summer warming on the thermal characteristics of a polymictic lake and consequences for oxygen, nutrients and phytoplankton. *Freshwater Biology*, *53*(2), 226-237. https://doi.org/10.1111/j.1365-2427.2007.01887.x
- Winder, M., Reuter, J. E., & Schladow, S. G. (2009). Lake warming favours small-sized planktonic diatom species. Proceedings of the Royal Society B: Biological Sciences, 276(1656), 427-435. https://doi.org/10.1098/rspb.2008.1200
- Gray, E., Mackay, E. B., Elliott, J. A., Folkard, A. M., & Jones, I. D. (2020). Wide-spread inconsistency in estimation of lake mixed depth impacts interpretation of limnological processes. Water Research, 168, 115136. https://doi.org/10.1016/j.watres.2019.115136
- Wilson, H. L., Ayala, A. I., Jones, I. D., Rolston, A., Pierson, D., de Eyto, E., ... & Jennings, E. (2020). Variability in epilimnion depth estimations in lakes. Hydrology and Earth System Sciences, 24(11), 5559-5577.

https://doi.org/10.5194/hess-24-5559-2020
- Chen, C. T. A., & Millero, F. J. (1986). Thermodynamic properties for natural waters covering only the limnological range 1. Limnology and Oceanography, 31(3), 657-662. https://doi.org/10.4319/lo.1986.31.3.0657
- Boehrer, B., & Schultze, M. (2008). Stratification of lakes. Reviews of Geophysics, 46(2). https://doi.org/10.1029/2006RG000210

*In conclusion, I find this contribution is not ready for publication:*
*while some of the shortcomings can be removed easily, a few things remain, which are hard to correct.*

We have addressed all the corrections requested by the referees and we think that our paper deserves publication for the novelty of the modelling approach in the context of volcanic lakes and for the deeper understanding it provides of Lake Albano's dynamics given that this work implements a 3D model able to describe the main features of this lake. Moreover, this study represents a necessary step toward further studies on the lake's evolution under future global warming scenarios (companion paper in preparation), as well as on the potential hazard impacts on the surrounding areas.

*smaller points:*
*line 42: not only volcanic: gas pressure of concern also through geochemical processes (Sanchez-Espana: Guadiana pit lake) or pollution (Horn et al.: Vollert pit).*

We mentioned the papers indicated by the referee on non-volcanic lakes, adding lines 45-46.

*line 141: add the charges to the ions.*

Done

*Figure 3: 0.1 g/kg of salinity difference is NOT homogeneous; This has even noticeable effect on density.*

We agree that 0.1 g/kg has a noticeable effect on density; we used the term "homogeneous" at line 134 improperly: we actually meant "spatially uniform", as detailed at lines 142-144. We removed the word "homogeneous" in the text at line 140 in the present version of the paper.

*line 153 why ERA 5 , was there no better source for weather data?*

The numerical experiment presented here serves as an initial benchmark for this novel model configuration, which we are using to perform much longer simulations in the past (to be described in the companion paper currently in preparation). ERA5 dataset ensures both high quality and long temporal coverage in the past, as mentioned at lines 161-162 of the manuscript. Therefore, we decided to test the model using a setup as close as possible—including the atmospheric forcing—to the one that will be used in the forthcoming long-term numerical experiments.

*Figure 6: coast ? -> shore line*

We agree that "shoreline" is more appropriate. We changed the legend in the figure.

*Figure 7: This comparison of temperature. I see a deviation between 1 and 5°C from the SE observation, also in winter: Why are the authors convinced that the model is good enough to simulate deep mixing?*

During the project we realized that the SE data logger (whose recorded temperature time series is shown in Fig. 7, blue curve) after January probably moved and was not in the water for an uncertain time window. This was not the case for the W data logger. For completeness we preferred to show all the recorded data and discuss the relative field problems, but considering this comment by the reviewer we decided to show only the part of the time series when the datalogger was certainly in the water. In our view, the W data logger which did not have similar problems is in rather good agreement with the model results. Also, we considered other data gathered in the time window 2020-2023. Unfortunately, the scarcity of data forced us to use other data from other points of the lake in locations far from the datalogger locations, in particular in the deepest point and in different sparse points in the lake (so an averaged value is shown). So they were pachy, non continuous, and in the majority of the cases were collected in the central part of the lake at specific depths. However, we believe that the model is able to reasonably reproduce the surface seasonal temperatures and their variability with an acceptable RMSE.

*line 246: different -> separate*

Done

*Figure 10, 11, 12 , 13: where is north or south: write clearly the*

*months over the columns and dates in front of rows and write in the
caption what is displayed.*

We agree that the x label has a mistake and we corrected it.

*Figure 15 has a bad colourbar for the temperature.*

We changed the Figure showing the density and using a different colorbar.

*line 343: what is the density approach that is used?*

We re-computed the Schmidt stability index (Figure 14 and equation 2) using the
density approach by Chen and Millero (1986), as suggested in Boehrer & Schultze
(2008) for low salinities (<0.6 psu).

**Reviewer RC2**

*In this study, a three-dimensional model is implemented to investigate the thermal
dynamics and mixing in Lake Albano, a crater lake with high accumulation of
dissolved CO2 and thus a certain potential for a limnic gas eruption.*
*The purpose of the present study is not sufficiently clearly defined. The text states "to
estimate the potential gas hazard of Lake Albano, numerical modeling of the lake
water dynamics is crucial for understanding its current and future behavior and
stability." and "In this study, we investigate the characteristics of lake stratification
and overturning events at Lake Albano through the results of 3D numerical model
simulations, ...". However, these are rather general statements, and there are no
specific research questions or hypotheses mentioned.*

In our plans, this is the first paper on the modelling of Lake Albano, and a companion
paper is in preparation to explore how the volcanic lake dynamics may change under
climate change scenarios. We acknowledge that some statements may currently
appear generic. However, this paper is intended as the initial part of a more
comprehensive study.
According to our intentions, the main goal of this paper was to describe the setup of
an initial framework (in terms of numerical tools, observations, and diagnostic
methods) aimed at investigating the characteristics of lake stratification and
overturning events. The final research question, which should probably be stated
more clearly in the manuscript, is whether the changes driven by global
warming—intensified in Mediterranean regions—might significantly inhibit the
preconditioning phases necessary for Lake Albano's overturning, thus facilitating

possible hazardous releases of $CO_2$.

*Predicting mixing in deep warm "monomictic" lakes is typically a challenging task for lake models. I write "monomictic" in quotation marks, because in many of these lakes, the seasonal mixing depth is variable depending on the meteorological conditions in winter, and only reaches the full depth of the lake every few years in cold winters. The lakes are thus often not truly monomictic. This occasional complete mixing results in a typical slow increase in hypolimnion temperature during years with incomplete mixing and a faster decline in temperature and a sudden increase in oxygen concentrations during complete mixing events. This is the case, for example, for many of the Northern Italian deep lakes (Rogora et al., 2018, 10.1007/s10750-018-3623-y), and based on previous data, it is probably also true for Lake Albano (Ellwood et al., 2009, doi:10.3274/JL09-68-2- 12). Whether or not a complete overturn occurs in a specific year may depend on relatively minor differences in atmospheric forcing, and mixing can also be inhibited by additional chemical density stratification due to salinity (typically resulting from organic matter mineralisation and/or calcite dissolution) and in the case of CO2-rich lakes such as Lake Albano also due to the density contribution of CO2.*

We agree that predicting mixing in deep, warm "monomictic" lakes can be a challenging task for lake models. The mixing dynamics we intend to investigate refer to deep mixing events that reach the $CO_2$-rich bottom layers of the lake (below 40 m depth) and are capable of transporting this $CO_2$-laden water toward the surface, allowing the gradual release of $CO_2$ into the atmosphere over time. The ultimate goal of this comprehensive study is to determine whether the regular release of $CO_2$ to the atmosphere may be compromised in the future, and whether $CO_2$ could become hazardous if released after accumulating over many years without regular overturning. This objective will be addressed in the aforementioned companion paper, while the present paper aims to lay the numerical and methodological foundations for the subsequent steps of this comprehensive study. We acknowledge that, without clearly stating our intentions in the manuscript, the research question and the approach used could appear unclear, we will therefore revise the manuscript to better clarify the overarching goals of this comprehensive study, of which the present paper represents the first step.

*That said, I am not convinced that the model presented here can be used to reliably predict mixing in Lake Albano for the following reasons:*

*•Most importantly, in my opinion, it does not make sense to prescribe a constant vertical turbulent diffusivity in a lake model if the goal of the model is to project vertical mixing. The turbulent diffusivity typically varies by several orders of*

*magnitude both vertically depending on forcing and stratification (with highest diffusivity in the surface layer, low diffusivity in the metalimnion and intermediate diffusivity in the hypolimnion) and seasonally. A correct representation of these dynamics in vertical diffusivity is required to reliably predict vertical mixing in a lake. As far as I know, the SHYFEM model does have the option to calculate vertical diffusivity using a range of turbulence closure approaches. Why was none of these options used?*

The numerical experiment mentioned in line 169 of the manuscript, with results presented in Fig. 4, was designed to test the vertical molecular diffusivity parameter for temperature and salinity in the SHYFEM model (referred to as the *difmol* parameter in the code). To account for turbulence, the model computes vertical eddy viscosity and diffusivity using the k-$\varepsilon$ turbulence closure scheme implemented in the GOTM (General Ocean Turbulence Model) framework, which is integrated into SHYFEM. We acknowledge that not explicitly stating that the tested parameter was the molecular diffusivity led to confusion. We will revise the manuscript accordingly to clarify this point and prevent any misunderstanding.

*•The model performance in predicting surface temperature is rather bad, with RMSE > 3°C overall and > 2°C in winter, and has a clear bias (Figure 7). A temperature difference of a few tenths of a degrees can determine whether the lake does or does not mix completely in winter. I think significant additional work would be required to better calibrate the heat flux parameterizations in the model to achieve better agreement with observed surface temperatures.*

As already mentioned in response to one of Referee #1's comments, during the project, we discovered that the SE data logger (whose temperature time series is shown as the blue curve in Fig. 7) was likely displaced and, for an unspecified period, may have been out of the water. This issue did not affect the W data logger which, in our view, is in rather good agreement with the model results. Unfortunately, the scarcity of data does not allow for particularly in-depth validation. However, we removed the data which was likely a measure of the air temperature and not water temperature and kept the initial data of the time series, when we are sure the datalogger was in the water lake. The re-computed RMSE improved as we reported in the manuscript. We believe the model is able to reasonably reproduce the main temperature structures according to the seasonal variations of the Lake Albano latitude.

*Potentially, also the ERA5 forcing data is not representative for the lake. Specifically, in this hilly region the elevation of the ERA5 grid cell could differ quite a bit from the lake's elevation and thus result in a bias in air temperature, and local wind speed*

*could significantly differ from the ERA5 grid (as also implied by the analysis of observed currents). It might be helpful to compare with observations from the local meteo station in Viterbo, just a few km south of the lake at similar elevation (https://oscar.wmo.int/surface/index.html#/search/station/stationReportDetails/0-2000 0-0- 16216).*

We agree that using atmospheric forcing from local observations could improve some aspects of the numerical results, as we mention with regard to surface circulation (see Fig. 11 in the present version of the paper). Nevertheless, as already mentioned earlier in this document, this numerical experiment represents an initial benchmark for the novel model configuration, which will also be applied to longer-term historical simulations, to be discussed in a forthcoming companion paper. This is the reason why the model was tested using a configuration as close as possible—atmospheric forcing included—to that planned for the upcoming long-term numerical experiments. In light of the reviewer's insightful and helpful comment, we carried out a comparison between the air temperature from the ERA5 dataset and that recorded at the Ciampino weather station (from the Global Historical Climatology Network Daily database), which is closer to Lake Albano than the suggested station in Viterbo.

In the figure below we report the daily mean air temperature from the ROMA CIAMPINO (IT000016239) weather station (blue curve) and from the closest grid point of the ERA5 dataset (red curve) over the period 2020–2023. The weather station data are derived from the GHCN (Global Historical Climatology Network) Daily database (Menne et al., 2012a; Menne et al., 2012b). The values in the upper-left corner of the plot indicate the mean Root Mean Square Error (RMSE) and the mean bias of the ERA5 dataset with respect to the weather station data, for the considered model grid point and over the considered period. In our opinion, the RMSE and the bias are within the observed temperature variability.

[Figure]

Additional references (not included in the paper):

- Menne, M.J., I. Durre, R.S. Vose, B.E. Gleason, and T.G. Houston, 2012a:

An overview of the Global Historical Climatology Network-Daily Database. Journal of Atmospheric and Oceanic Technology, 29, 897-910, doi:10.1175/JTECH-D-11-00103.1.
- Menne, M.J., I. Durre, B. Korzeniewski, S. McNeill, K. Thomas, X. Yin, S. Anthony, R. Ray, R.S. Vose, B.E.Gleason, and T.G. Houston, 2012b: Global Historical Climatology Network - Daily (GHCN-Daily), Version 3.32 NOAA National Climatic Data Center. http://doi.org/10.7289/V5D21VHZ [access date 09 Jul 2025]

*•There is no evaluation whether the model is actually able to correctly reproduce previous observed mixing events.*

Unfortunately, for the period we simulated (2020–2023), there are, to our knowledge, no specific studies on overturning events in Lake Albano. Since model validation was a crucial first step, we chose to simulate this period because it corresponds to the timeframe during which we collected field data.
The hypothesis that the Lake Albano overturning may be modified in the next decades by global warming will be investigated in the companion paper in preparation.

*Furthermore, I disagree with the assessment that salinity is not important for density stratification and mixing in Lake Albano. The annual mean profile presented in Figure 3 shows a salinity difference of 0.1 g/kg between the surface and the bottom of the lake. This approximately corresponds to a density difference of 0.08 g/kg (see e.g., Boehrer and Schultze, 2008, doi: 10.1029/2006RG000210). This is equal to the density difference between water at 9 °C and water at 8 °C. Thus, if the deep water of Lake Albano has a temperature of 9°C, the surface water would have to cool down to 8°C to reach the density of the deep water and allow mixing. The analysis of mixing depth purely based on temperature differences in section 3.4 is therefore in my opinion not valid. This also means that a model would likely need to adequately reproduce the sources of salinity in Lake Albano to correctly predict mixing in years beyond the first year where salinity may by prescribed by the initial conditions.*

We did not write that salinity is not important for the density stratification and we added Figure 9 showing the agreement between the observed background observation and other data gathered during the project. Following both reviewers' comment we recomputed the analysis of the mixing depth using 3 methods:Temperature Threshold Method; Density Threshold Method; and Max Temperature Gradient. We substitute Figure 13 to the manuscript showing the Howmoller diagram of the density profiles at the deepest point of the lake and the mixed layer depth. By comparing the mixed layer depth computed using the Density

Threshold Method and the concomitant work on the observed CO2 degassing in the time window 2019-2024 (Rouwet et al, 2025 in preparation) we found that the overturning events occurred in 2021, 2022 and 2023 winters (not in 2020 winter)

*Finally, it would be useful to add some discussion on the model choice. There are several widely tested 1D lake models available to simulate vertical mixing in lakes (eg GLM, Simstrat), and the simple morphology of Lake Albano with a single deep basin seems to be an ideal case for applying such a model. I do not really see anything in the present manuscript that would justify applying a computationally far more expensive 3D model instead, which is less tested in the context of the present study to simulate vertical mixing in lakes. The manuscript does include some discussion of the spatial dynamics of mixing, which typically starts near the shore due to faster local cooling in shallower parts of the lake. However, this is a known and well-studied effect (eg. Boufard et al., 2025, doi: 10.1017/fo.2024.31), and there is no discussion in the manuscript why this spatial variability should be relevant for investigating vertical mixing in the context of potentially dangerous CO2 emissions.*

We aimed to design a tool capable of providing broader information on a lake system that has been the subject of various observational studies but, to our knowledge, has never been addressed through modelling. Therefore, we considered it more appropriate, as a first modelling approach to the study area, to develop a tool applicable to other aspects of the system beyond mixing and overturning dynamics. A 1D model could prove advantageous for future studies, once a solid understanding of the vertical and horizontal dynamics of the study area has been achieved.

*Some details:*
*•The title of the manuscript is rather unspecific and could refer to any type of dynamics in the lake.*

We preferred to keep the title generic because this study serves as an initial step in the numerical investigation of Lake Albano, with the observational component being of crucial importance in supporting the model validation.

*•I think in equation (20), there should be division, not multiplication with A0. The numbers in the figure look reasonable, though, so I assume this was calculated correctly.*

We checked the code, we divided by A0, but the formula was written wrongly in the manuscript and we corrected it.